# Complete Genome Analysis of *Rhodococcus opacus* S8 Capable of Degrading Alkanes and Producing Biosurfactant Reveals Its Genetic Adaptation for Crude Oil Decomposition

**DOI:** 10.3390/microorganisms10061172

**Published:** 2022-06-07

**Authors:** Yanina Delegan, Kirill Petrikov, Ekaterina Frantsuzova, Natalia Rudenko, Viktor Solomentsev, Nataliya Suzina, Vasili Travkin, Inna P. Solyanikova

**Affiliations:** 1Laboratory of Physiology of Microorganisms, Institute of Biochemistry and Physiology of Microorganisms, Federal Research Center “Pushchino Scientific Center for Biological Research of Russian Academy of Sciences” (FRC PSCBR RAS), 142290 Pushchino, Moscow Region, Russia; mewgia@yandex.ru (Y.D.); katkaty235@gmail.com (E.F.); 2State Research Center for Applied Microbiology and Biotechnology, 142279 Obolensk, Moscow Region, Russia; solomentsev@obolensk.org; 3Laboratory of Plasmid Biology, Institute of Biochemistry and Physiology of Microorganisms, Federal Research Center “Pushchino Scientific Center for Biological Research of Russian Academy of Sciences” (FRC PSCBR RAS), 142290 Pushchino, Moscow Region, Russia; bioscience.kp@gmail.com; 4The Federal State Budget Educational Institution of Higher Education Pushchino State Institute of Natural Science, 142290 Pushchino, Moscow Region, Russia; 5Institute of Basic Biological Problems of Russian Academy of Sciences, Federal Research Center “Pushchino Scientific Center for Biological Research of Russian Academy of Sciences” (FRC PSCBR RAS), 142290 Pushchino, Moscow Region, Russia; nataliacherry413@gmail.com; 6Laboratory of Cytology of Microorganisms, Institute of Biochemistry and Physiology of Microorganisms, Federal Research Center “Pushchino Scientific Center for Biological Research of Russian Academy of Sciences” (FRC PSCBR RAS), 142290 Pushchino, Moscow Region, Russia; suzina_nataliya@rambler.ru; 7Regional Microbiological Center, Belgorod National Research University, 308015 Belgorod, Belgorod Region, Russia; travkin-55@mail.ru; 8Laboratory of Microbial Enzymology, Institute of Biochemistry and Physiology of Microorganisms, Federal Research Center “Pushchino Scientific Center for Biological Research of Russian Academy of Sciences” (FRC PSCBR RAS), 142290 Pushchino, Moscow Region, Russia

**Keywords:** genome sequencing, *Rhodococcus opacus*, crude oil, degradation, soils, bioremediation, cold-adaptation

## Abstract

Microorganisms capable of decomposing hydrophobic substrates in cold climates are of considerable interest both in terms of studying adaptive reactions to low temperatures and in terms of their application in biotechnologies for cleaning up oil spills in a crude-oil polluted soil. The aim of this work was to investigate the genome of *Rhodococcus opacus* S8 and explore behavior traits of this strain grown in the presence of hexadecane. The genome size of strain S8 is 8.78 Mb, of which the chromosome size is 7.75 Mb. The S8 strain contains 2 circular plasmids of 135 kb and 105 kb and a linear plasmid with a size of 788 kb. The analysis of the genome revealed the presence of genes responsible for the degradation of alkanes and synthesis of biosurfactants. The peculiarities of morphology of microbial cells when interacting with a hydrophobic substrate were revealed. An adaptive mechanism responsible in the absence of oxygen for maintaining the process of degradation of hexadecane is discussed. The data obtained show that the strain S8 has great potential to be used in biotechnologies.

## 1. Introduction

Pollution of the environment by crude oil and oil products is a global problem of our time [1]. The main characteristics of this type of pollution include its organogenic nature, multicomponent composition, the toxicity of components, the scale of distribution, the high migration ability and resistance of heavy fractions to decomposition, hydrophobicity of compounds and ability of these compounds to migrate through food chains [2]. Problems that arise during oil production, its transportation, storage, as well as natural oil leakage from the places of its formation lead to serious environmental pollution with hydrocarbon substrates. Oil spills damage the environment in a variety of way all over the world and can have negative impacts on natural ecosystems [3]. In biotechnologies of cleaning up oil spills from contaminated areas, an important place is given to microorganisms, the vast majority of which belong to the genus *Rhodococcus* [4,5]. The ability to use even low concentrations of organic substrates, halotolerance, the ability to grow at low or, conversely, high temperatures are the characteristic features of these bacteria.

In soil samples taken from the arctic-subarctic zone, rhodococci and other actinobacteria dominated in soil with a low content of organic matter [6]. All these properties are important when microorganisms are used in bioremediation processes, including bioaugmentation, i.e., highly effective bacteria are added to stimulate biodegradation processes [7]. Strategies used by microorganisms to degrade oil hydrocarbons include synthesis of appropriate enzymes, cometabolism, the transfer of catabolic plasmids, and the production of biosurfactants to increase the bioavailability of hydrophobic substrates [8]. There are many works devoted to the study of the genetic control of the processes of biodegradation of various hydrocarbons, which demonstrate the increased interest of researchers in understanding the mechanisms of biodegradation [9,10,11].

According to the degree of reduction of oxidation by microorganisms, the components of oil and oil products are arranged in the following sequence: *n*-alkanes → branched alkanes → branched alkenes → low molecular weight *n*-alkyl aromatic compounds → monoaromatic compounds → polycyclic aromatic hydrocarbons → asphaltenes [12,13,14]. Temperature plays a significant role in the course of all chemical reactions, including those realized by microorganisms. As known, temperature affects the solubility of compounds, their physical state, and this is directly related to the bioavailability of components for microbial cells. The ambient temperature affects the state of the cell, the properties of its membranes and the activity of enzymes. The efficiency of bioremediation at low temperatures has been little studied to date, and the problem of cleaning up oil spills has not yet been solved. Thus, the ability of four actinobacteria adapted to a cold climate to decompose some oil components at a low temperature, from 1 to 20 °C, was studied [15]. *R. erythropolis* strain BZ4 was reported to be able to degrade C12-C22 *n*-alkanes, phenol and polyaromatic hydrocarbons during cultivation at 1–30 °C. *R. cercidiphyllus* BZ22 at 1–20 °C fully degraded C12 (700 mg/L), the degradation of C16 was less promoted. The psychrotolerant *Rhodococcus* sp. Y2-2 strain also showed the ability to decompose crude oil components at low temperatures. 84% of mixture presented by 500 mg/L of each kerosene, gasoline, and diesel fuel, was degraded by *Rhodococcus* sp. Y2-2 after 2 weeks of cultivation at 10 °C. The addition of glucose stimulated the degradation process. The strain under the same cultivation conditions decomposed 4 g/L of the mixture. The degradation process was accompanied with the synthesis of surfactants [16].

Thus, microorganisms capable of biosynthesizing surfactants at low temperatures, synthesizing microbial enzymes responsible for degradation, transporting hydrophobic compounds into the cell, and adapting to the presence of elevated salt concentrations are of great interest from both theoretical and practical points of view. Therefore, it is especially important to study the processes of biodegradation and bioremediation, as well as the development of effective biological products and technologies for cleaning up oil spills in cold and temperate areas.

The aim of this work was to select the most effective strain among *Rhodococcus* spp., capable of reducing surface tension during growth in the presence of hexadecane, to characterize it by genome sequencing and biosurfactant isolation and to formulate a hypothesis for the provision of oxygen to cells during the interaction of a cell and a hydrophobic substrate. Among 13 strains, *R. opacus* S8 was selected for its ability to reduce surface tension when cultivated on hexadecane.

## 2. Material and Methods

### 2.1. Bacterial Strains and Cultivation Conditions

The strains used in this work were previously isolated from sites contaminated with various pollutants, including oil products, and were stored in the collection of the plasmid biology laboratory and in the collection of the laboratory of the enzymatic degradation of organic compounds (IBPM RAS (FRC “PSCBR” RAS), Pushchino, Russia).

The strains were cultivated in Evans mineral medium (per 1 L): K_2_HPO_4_-8.71 g; 5 M NH_4_Cl solution-1 mL; 0.1 M Na_2_SO_4_ solution-1 mL; 62 mM MgCl_2_ solution-1 mL; 1 mM CaCl_2_ solution-1 mL; 0.005 mM solution of (NH_4_)_6_Mo_0_O_24_-1 mL; microelement solution-1 mL; HCl conc.-up to pH = 7.5. The composition of the solution of trace elements in 1% HCl solution is as follows in g/L: ZnO-0.41; FeCl_3_ 6H_2_O-5.4; MnCl_2_ 4H_2_O-2; CuCl_2_ 2H_2_O-0.17; CoCl_2_ 6H_2_O-0.48; H_3_BO_3_-0.06. Hexadecane was used as carbon-energy source (2% *v*/*v*).

### 2.2. Determination of the Surface Tension

To assess the surface activity, the microorganism was grown on a modified Evans medium with hexadecane as carbon-energy source. Cultivation was carried out in Erlenmeyer flasks on an orbital shaker (200 rpm) at 28 °C for 4 days.

To measure the surface tension, 50 mL of the culture liquid was transferred into a separating funnel. The lower aqueous phase without hexadecane layer was collected. The measurement was carried out by the Du Noüy ring method on a K6 tensiometer (Kruss, Hamburg, Germany) at room temperature. The surface tension of a pure Evans medium was 72 mN/m.

### 2.3. Whole Genome Sequencing

To obtain complete genomic assembly, genomic DNA sequencing was performed using Illumina and Oxford Nanopore technologies. The Monarch HMW DNA Extraction Kit enabled fast isolation of genomic DNA from the *Rhodococcus opacus* S8 strain. The concentration and quality of the resulting preparation was evaluated on a Qubit Fluorometer (Thermo Fisher Scientific, Waltham, MA, USA). Nanopore sequencing was performed using MinION equipment with an R9.4.1 flow cell (Oxford Nanopore Technologies [ONT], Oxford, UK). The Ligation Sequencing kit SQK-LSK109-K was used to prepare the libraries, and the Guppy program was used for the base calling procedure. 2,158,739,097 nucleotides were obtained, distributed over 1,282,908 reads (N50 7.329). Quality Q > 10 corresponded to 986,194 reads (1798.1 Mb of data)-76.9% of the original number of reads. Raw nanopore reads were filtered by quality (Q > 10) and length (l > 2000) using the NanoFilt program from the NanoPack package (ver. 2.8.0, Wouter De Coster et al., Antwerp, Belgium) [17].

Whole genome sequencing of strain S8 using Illumina technology was performed on Illumina NovaSeq 6000 platform equipment using S2 reagent kit (catalog number 20012861; 2 × 100 bp) at BioSpark (Troitsk, Moscow), libraries were prepared using KAPA HyperPlus Kit (KAPAbiosystems, Wilmington, MA, USA).

The FastQC program (http://www.bioinformatics.babraham.ac.uk/projects/fastqc, accessed on 1 August 2019, ver. 0.11.9) was used to control the quality of Illumina raw reads. Illumina raw reads were filtered: adapter sequences, low quality reads, and short reads were removed. Reads obtained using Illumina and Oxford Nanopore technologies were used for hybrid assembly by the SPAdes program [18]. Separately, nanopore reads were assembled into contigs using the Flye program [19]. The SPAdes contigs were then assembled into replicons using the Flye results. Illumina reads were used to fix minor build errors with Bowtie [20] and Pilon [21] programs. Circularization of the ends of replicons (chromosomes and plasmids) was confirmed by end-overlapping as well as visualization in the Tablet program [22].

### 2.4. Estimation of Expression of Target Genes

The *R. opacus* S8 strain was cultivated in two conditions: Evans mineral medium + glucose 10 g/L (control) and Evans mineral medium + hexadecane 2% *v*/*v* (experiment).

Evans mineral medium + hexadecane 2% *v*/*v* was chosen to assess the surface activity of the strains and the production of surfactant preparations. The total RNA of the strains was isolated from the biomass obtained by centrifugation from the culture liquid. The total RNA of the strains was obtained using the Aurum total RNA mini kit (Bio-Rad, Hercules, CA, USA), the RNA concentration was determined using Nanodrop equipment (Thermo Fisher Scientific), and the quality of the preparations was assessed by agarose gel electrophoresis. The RevertAid RT Reverse Transcription Kit (Thermo Fisher Scientific) provided a procedure for synthesis of the first cDNA strand. All experiments were performed in 5 biological replicates; the 16S rRNA gene was used as a reference gene. Quantitative reverse transcription polymerase chain reaction was performed with qPCRmix-HS SYBR (Evrogen, Moscow, Russia). qRT-PCR data were normalized against 16S rRNA gene. PCR reactions were carried out in LightCycler 96 Instrument, Roche Diagnostics GmbH (Mannheim, Germany).

Statistical data processing was performed in the RStudio program, the difference between the mean values of the samples was determined using Student’s *t*-test for two independent samples. The primer sequences for the genes of interest are presented in Table 1. Amplification efficiency was determined using a series of tenfold dilutions of the DNA template, the specificity of the reaction was confirmed by agarose gel electrophoresis.

### 2.5. Isolation of Biosurfactants

To isolate biosurfactants, the microorganisms were cultivated as described above. The biomass was precipitated by centrifugation for 40 min at 4200 rpm on a J6-M1 centrifuge (Beckman Coulter, Brea, CA, USA) at 4 °C. The resulting cell-free supernatant was acidified with concentrated hydrochloric acid to pH = 2. The solution was left overnight at 4 °C. Then extraction was carried out with methyl tert-butyl ether in a volume ratio of 1:1. The upper organic layer was collected; the solvent was removed on a rotary evaporator. The resulting preparation was purified on a 20 × 1 cm glass chromatographic column. Silica gel 60, 70–230 mesh (Merck, Darmstadt, Germany) was used as a sorbent. Elution was carried out first with chloroform to remove hexadecane impurities, then with a mixture of chloroform:methanol (10:2 *v*/*v*).

Identification of glycolipids in the obtained fractions was carried out by thin layer chromatography on silica gel 60 plates (Merck), eluting with a mixture of chloroform:methanol:water (65:15:2 *v*/*v*/*v*). The plates were treated with a naphthol reagent (0.5 g of α-naphthol in 100 mL of methanol-water, 1:1 *v*/*v*), then with 10% sulfuric acid, and heated for 2 min at 110 °C. Glycolipids appeared as purple spots.

### 2.6. Microscopy

#### 2.6.1. Phase Contrast Microscopy

For light microscopy, a Zeiss Axio Imager A1 microscope (Zeiss, Oberkochen, Germany), an Axiocam 506C camera, and a Zeiss 56HE filter set (excitation: 470/27, emission: DBP 512/30 + 630/98) was used in this study.

#### 2.6.2. Electron Microscopy

Cells were concentrated by centrifugation (10,000× *g*, 15 min), fixed with 2% glutaraldehyde in 0.05 M cacodylate buffer (pH 7.2) for 1 h at 4 °C and further treated as described earlier [23]. Ultrathin sections were viewed under a JEM-1400 transmission electron microscope (JEOL, Musashino, Japan) at 80 kV.

## 3. Results and Discussion

### 3.1. Surface Tension

A collection of rhodococci was screened for the ability to synthesize biosurfactants during growth on hexadecane. Estimation of surface tension (ST) is the most reliable method for detecting surface-active compounds, and a decrease in the value of ST down to 40 mN/m and below is the criterion for the effective production of biosurfactants [24]. All strains used in this study were previously isolated from soil and water contaminated with hydrocarbons or (chloro)biphenyls. These substances are hydrophobic, for the effective decomposition of which their emulsification is necessary [25]. Our findings demonstrate that the strains S8 and VT-6 decrease surface tension to values that indicate a potentially high surfactant-producing activity (Table 2). It can be noted that a comparable proportion of biosurfactant producers is often detected during screening of bacterial isolates from natural samples. For example, Bodor et al., (2003) showed that only 4% of strains (45 out of 1305) isolated from soil samples in dry regions of the United States can reduce surface tension to the level indicated above [26]. The best results reported by these authors are 27.3 ± 0.3 mN/m and 28.7 ± 0.4 mN/m for *Pseudomonas* spp. For rhodococci, there is evidence that a number of strains are capable of reducing surface tension to significant values. Thus, *Rhodococcus* sp. LF-13 and *Rhodococcus* sp. LF-22 reduced the value of surface tension during growth on kerosene, *n*-hexadecane or rapeseed oil to values of 27–36 mN/m depending on growth substrate [27]. Strain *Rhodococcus* sp. TW53 isolated from deep-sea sediment decreased the surface tension down to 34 mN/m [28].

Whole genome sequencing of the VT-6 strain has been previously reported [29]. Therefore, in this work, we paid special attention to the ability of S8 to synthesize biosurfactants.

### 3.2. Genome Sequencing

S8 was identified as *R. opacus* based on sequence analysis of the 16S rRNA gene fragment. When sequencing the genome of this strain using Illumina technology, 23,607,586 reads with a total length of 2,328,015,660 nucleotides were obtained. Duplicates, low quality and short reads for a total of 513,302 reads (4.45% of total reads) were removed. For further analysis, 22,023,056 reads were taken (95.55% of the total number of reads).

The total genome size of strain S8 is 8.78 Mb, of which the chromosome size is 7.75 Mb. The S8 strain contains 2 circular plasmids pCP135 of 135 kb and pCP105 105 kb. The strain also contains a linear plasmid pLPS8 with a size of 788 kb. The genome of the strain contains 4 rRNA clusters, 50 tRNAs, and 3 ncRNAs. There are 8215 genes in total, of which 8150 are CDS. The genome of the strain is deposited in the Genbank NCBI database under the numbers:Chromosome-CP093380pLPS8-CP093381pCP105-CP093383pCP135-CP093382 (Figure 1).

*R. opacus* chromosomes are, on average, about 1 Mb longer than *R. erythropolis* chromosomes (Table 3). It is interesting to note that, in contrast with *R. erythropolis*, alkane degrading *R. opacus* strains are not characterized by the maintenance of several copies of alkane monooxygenases in the genomes. Thus, the genome of S8 strain contains a single copy of *alkB*, accompanied by 2 copies of rubredoxin and rubredoxin reductase.

The *R. opacus* genomes have features typical of rhodococci: a complex structure with a large number of transposable elements, extrachromosomal elements (plasmids) several hundred kb in size. Notably, small plasmids (<100 kb) also occur, but there is no reported example of *R. opacus* strain where the small plasmids are present without megaplasmids. The *R. opacus* XM24D strain does not contain plasmids but has 2 circular chromosomes.

It is known that many bacteria with unstable plastic genomes (including representatives of the genus *Rhodococcus*) use plasmids as a repository of genes that are not constantly necessary in cell metabolism [30]. Therefore, plasmids often contain genes for the catabolism of different pollutants. Upon functional annotation of the pLPS8 plasmid, we found catabolic genes for various compounds, in particular:naphthalene catabolism gene clustercatechol 2,3-dioxygenase (EC 1.13.11.2)phthalate catabolism genes3 copies of cytochrome P450 hydroxylaseThe presence of truncated hemoglobin genes in the genome of the studied *R. opacus* S8 should be noted. The strain *R. opacus* S8 has 2 genes located on the chromosome. On the contrary, similar genes were not found in the genome of the *R. qingshengii* strain VT6.

Among the *R. opacus* plasmids, the pLPS8 plasmid is most closely related to the pROB02 plasmid of the *R. opacus* B4 strain (Figure 2). However, the pROB02 plasmid is much smaller (245 kb), while pLPS8 is 788 kb.

Plasmid pCP135 mainly consists of genes encoding proteins of unknown function (hypothetical). We found the Par plasmid division system in pCP135, as well as type II restriction-modification system genes and several transposase genes. A number of catabolic genes were found on plasmid pCP105, namely cyclohexanone monooxygenase (EC 1.14.13.22) and oxidoreductase, a short-chain dehydrogenase/reductase family. In addition, genes for restriction-modification systems and genes necessary for maintaining the plasmids themselves have been found.

All genes potentially involved in surfactant biosynthesis are located on the chromosome. The set of trehalose biosynthesis and transport genes in S8 is standard for *Rhodococcus* and includes:Trehalose 2-sulfotransferaseTrehalose-2-sulfate acyltransferase papA2 (2 copies)Acyltransferase papA3.

### 3.3. Estimation of Expression of Target Genes

When cells of strain S8 were cultivated on hexadecane, a significant (*p*-value < 0.01) increase in the amount of mRNA of the *alkB* gene was found (Figure 3). The expression of other genes did not depend on the carbon source.

### 3.4. Biosurfactants

It was found that strain S8 (Figure 4) forms two glycolipid products, Rf 0.55 and Rf 0.65. Most of the rhodococcal biosurfactants described in the literature are glycolipids [31], and the formation of several homologous products at once is typical. The formation of glycolipids in the VT6 strain was not detected. In the light of it, compounds of other types, possibly of a lipopeptide nature, are supposed to be responsible for the surface activity [28]. It was established that crude biosurfactants of *Rhodococcus* sp. TW53 were lipopeptides with five types of fatty acids in the structure [28].

### 3.5. Study of the Cell Morphology when Interacting with a Hydrophobic Substrate

The specific morphological characteristics of *Rhodococcus* cells were explored during growth on a hydrophobic substrate. Microbial cells grown on a mineral medium with glucose were used as a control. Under these conditions of cultivation, the cells have the shape of long sticks of uneven thickness, often branching, with characteristic large refractory inclusions, the size of which often exceeds the thickness of the cells (Figure 5). It can be assumed that cells under these conditions store triacylglycerides, which modify the shape of the cells.

Cultivation of cells in a rich medium for a long time showed that when cells reach a certain age and size, the process of crushing of rod-shaped cells occurs with the formation of chains of small cocci (Figure 6a). Figure 6b represents the formation of multiple partitions inside the rod-shaped cell at the initial stages of crushing. Refractory inclusions are reduced, although the determination of their presence in cells is difficult since the cells themselves are more refractive than cells grown in a mineral medium with glucose as a growth substrate.

When studying the morphology of cells cultivated in the presence of a hydrophobic substrate, it was found that the cells retain the shape of rods, which indicates active vegetative growth in the presence of a non-toxic substrate. Sorption of the initially hydrophobic substrate (hexadecane) on the cell surface is observed when bacterial cells are “pushed out” from the growth medium on hexadecane during preparation for light microscopy (applying the surface foam material, where the microbial utilization of hexadecane occurs, on a glass slide and pressing the drop with a cover slip) (Figure 7).

Cells of strain S8 are observed inside a drop with a growth medium on hexadecane. The arrows indicate cell conglomerates around the bubbles, which most likely represent air bubbles in the surface foam (Figure 8). This is an important observation, since rhodococci are aerobic bacteria that require the presence of oxygen in the growth medium. Adhesion of hexadecane droplets over the entire surface of the cells can probably hinder the diffusion of oxygen into the cells. Therefore, the observed non-uniform substance, “foam”, which can result from the dispersion of a hydrophobic substrate with the help of surfactants synthesized by bacteria, is a culture’s way of providing itself with oxygen access.

The cells of the *R. qinqshengii* VT6 strain, when cultivated on a rich medium, have a regular rod-like shape, which is also characteristic of other representatives of this genus. With an increase in cultivation time in a rich medium, the cell morphology does not change, although there is an uneven division of rod-shaped cells or their crushing into smaller cell forms (Figure 9).

When cultivating *R. qingshengii* strain VT6 inside a drop with growth medium on hexadecane, cells are exclusively localized inside fragments of a light amorphous substance (Figure 10).

One can see the cells inside the crushed drop with the growth medium on hexadecane (Figure 11) and the cells that were pushed out of the medium when pressed with a coverslip during preparation (Figure 11). The cells are predominantly in the form of thick rods of medium length.

Bacterial microconglomerates formed around a certain central core with refractory properties, which could be observed in phase contrast under conditions of growth on dodecane and hexadecane, appear to be organized specialized structures (Figure 8). In these structures, the refractory nucleus is an air bubble formed at the “liquid-air” interface with the participation of biogenic surfactants synthesized by the bacteria themselves. The location of the bacteria around the air bubble allows the bacteria to obtain the oxygen necessary for the degradation of alkanes. The extended fibrillar surface structures observed on ultrathin sections allow the bacterial cell to attach itself to the outer surface of the air bubble (Figure 12). Thus, bacterial cells are in a metabolically advantageous immobilized state with optimal access to both oxygen and substrate.

Every year a huge number of bacteria capable of destroying oil and oil products are isolated from various sources and their morphology, biochemistry, pigment synthesis, and genetic organization are studied [32,33]. Functional annotation of the *Rhodococcus erythropolis* strain PR4 genome, conducted by Laczi et al. made it possible to identify groups of genes whose up or down regulation gave a deeper understanding of molecular mechanisms involved in the degradation of hydrophobic substrates by bacteria [34]. In this regard, of great interest is the study of the interaction of cells with alkane-aqueous interface that occurs during the initiation of biofilm development by the bacterium *Marinobacter hydrocarbonoclasticus* SP17 when interacting with alkanes. The authors studied the adsorption of cells and found that after the cells reached the hexadecane-water interface as a result of diffusion process, they began to release surface-active compounds, which led to the formation of interfacial viscoelastic biofilms [35]. Contact between a glycolipid secreted by a strain of *Halomonas neptunia* isolated in Antarctica and *n*-hexadecane split into nanodroplets at the water interface was detected using transmission electron microscopy [36]. The authors found the formation of material with mesophase (liquid crystal) organization [36]. Studying the features of the synthesis of the rhamnolipid biosurfactant by the bacterium *Pseudomonas aeruginose*, Cameotra and Singh [37] found that the contact between the cell and the substrate occurred as a result of the formation of an emulsion under the action of the biosurfactant. The authors referred to this process as “internalization” of “biosurfactant layered hydrocarbon droplet” and suggested a mechanism similar in appearance to active pinocytosis, involving the internalization of a biosurfactant-coated hydrocarbon into the cell for subsequent degradation [37]. Hua et al. studying the growth of *Enterobacter cloacae* strain TU in a medium with hexadecane, found that this bacterium extracellularly released an exopolysaccharide (EPS) exhibiting bioemulsifying activity into the surrounding medium [38]. This EPS neutralized the ζ-potential of *E. cloacae* TU cells and increased surface hydrophobicity of cells, which favored the bioavailability of *n*-hexadecane. Important for understanding the processes of providing aerobic bacteria in contact with a hydrophobic substrate is the question of the formation of so-called gas balloons associating with a hemoglobin-like substance [39,40]. Oxygen transfer in such biofilms occurs with the participation of bacterial hemoglobin. The genes for flavohemoglobins were not found in the genome of strain *R. opacus* S8, however, two genes were presented on the chromosome, homologous to the genes encoding hemoglobin of other bacteria. They belong to group II and III truncated hemoglobin. The truncated hemoglobins are widely distributed in bacteria. It is assumed that one of their functions is the transfer of oxygen [41,42]. Thus, based on the available literature data and the results obtained in this work, it can be assumed that in the process of interaction of *Rhodococcus* cells with a hydrophobic substrate, many genes are involved, not only in the process of biodestruction, but also in the formation of specific extracellular structures that facilitate the interaction of cells with the substrate and, no less important, ensure the availability of oxygen for cells. There is practically no information on the cytological features of cell behavior when interacting with a hydrophobic substrate. The role of surfactants in the delivery of individual substrate molecules and the subtle mechanisms of cell interaction with organic substrate and oxygen have yet to be studied.

## 4. Conclusions

The ability of microorganisms to degrade crude oil and its individual components is very important in terms of developing technologies for biological methods of environmental cleanup. Among bacteria, representatives of the genus Rhodococcus are the most promising strains due to the complex organization of hereditary information, the presence of genes for the degradation of various organic compounds, the ability to synthesize biosurfactants and survive in a wide range of temperatures and adopt at a low content of substrates in the environment. The complete genome of 8.72 Mb in size was represented by a chromosome with a size of 8.45 Mb, two circular plasmids and one linear plasmid 788 kb. Genes and gene clusters involved in the degradation of hexadecane, surfactant biosynthesis, and adaptation to low temperatures were revealed. Due to the adaptive reaction of bacteria to growth in the presence of a hydrophobic substrate, morphological studies were undertaken to investigate the characteristics of cells grown on a mineral medium in the presence of hexadecane in comparison with those of cells grown on a mineral medium in the presence of glucose. An adaptive mechanism responsible in the absence of oxygen for maintaining the process of degradation of hexadecane was identified. The mechanism is based on the formation of microconglomerates of cells containing small volumes of air. Microconglomerates are formed as a result of physical processes of mixing and “swallowing” air from near-surface layers with the participation of surfactants of biogenic origin. Along with the discovery in the Rhodococcus genome of both groups of genes, alkane destruction genes and genes encoding truncated hemoglobin, this observation can become the basis for understanding the mechanisms of oxygen supply to bacterial cells for the aerobic destruction of hydrophobic substances. The data obtained show that S8 as a candidate capable of utilizing alkanes has great potential to be used in biotechnologies.

## Figures and Tables

**Figure 1 microorganisms-10-01172-f001:**
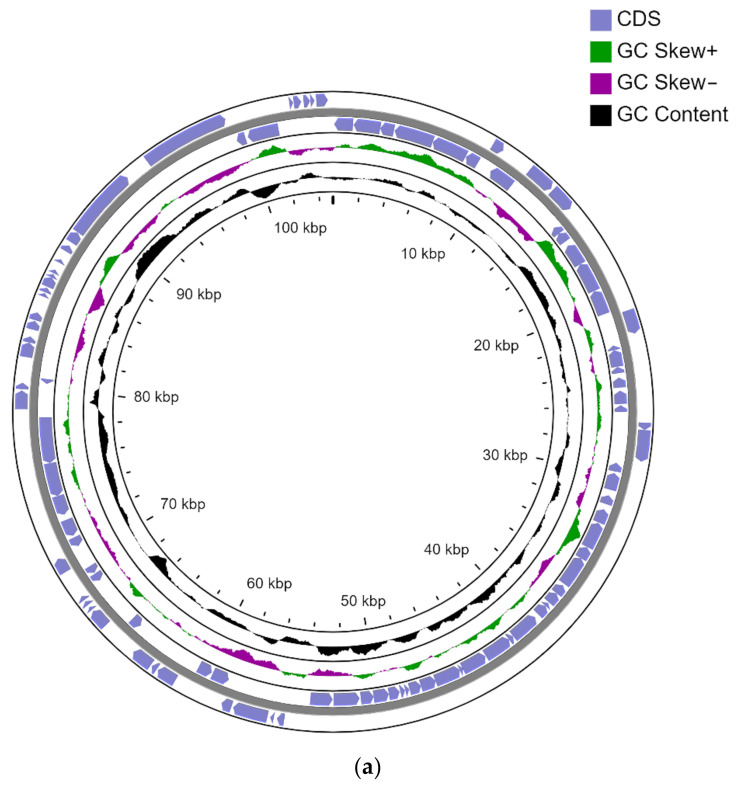
Circle maps of plasmids (**a**) pCP105 and (**b**) pCP135. From outside to the center: all CDS and RNA genes on forward strand, all CDS and RNA genes on reverse strand, GC content, and GC skew.

**Figure 2 microorganisms-10-01172-f002:**
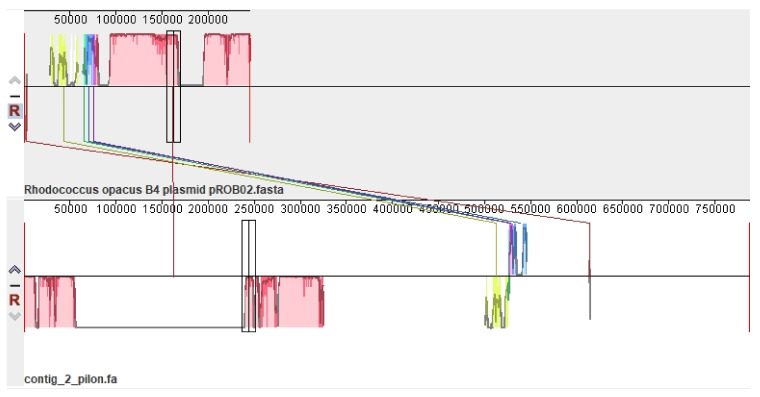
Mauve visualization of locally collinear blocks identified between pLPS8 of *R. opacus* S8 and its closest relative pROB02 of *R. opacus* B4. Vertical bars mark interchromosomal boundaries.

**Figure 3 microorganisms-10-01172-f003:**
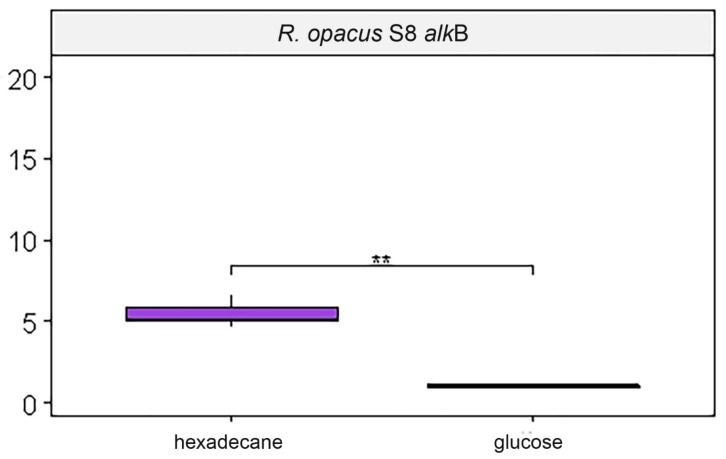
Real-time PCR. The relative amount of mRNA genes. ** *p* ≤ 0.01.

**Figure 4 microorganisms-10-01172-f004:**
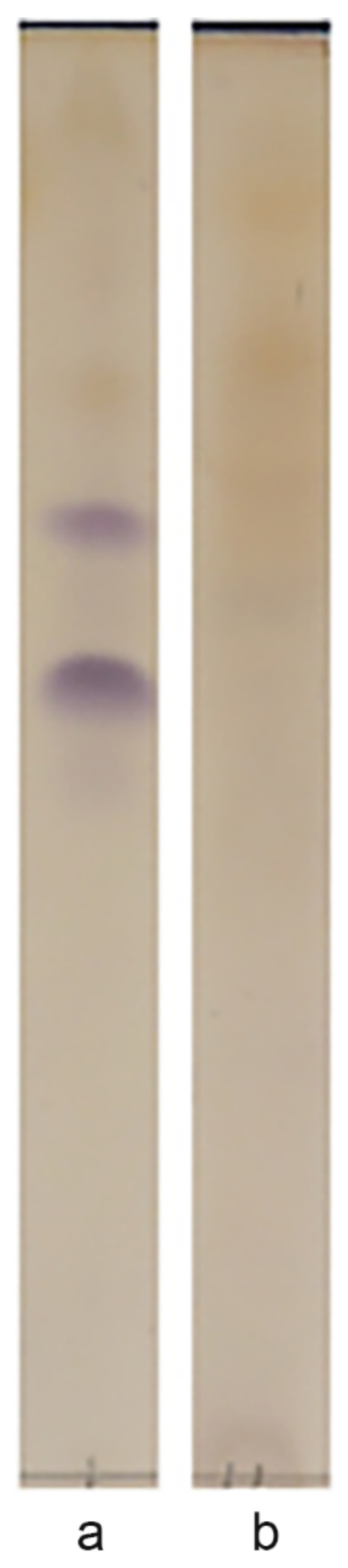
Chromatograms of glycolipids formed by (**a**) S8 and (**b**) VT6 strains.

**Figure 5 microorganisms-10-01172-f005:**
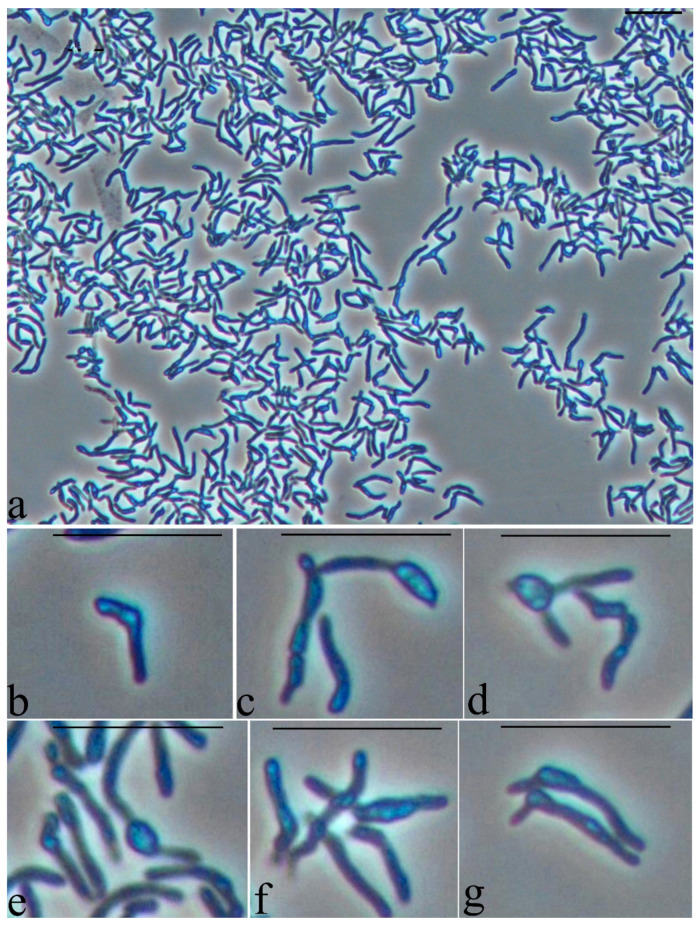
Strain S8 on glucose 1 day of growth. Light microscopy, phase contrast. The length of the scale bar is 10 µm. The cells have the shape of long sticks of uneven thickness, often branching (**a**,**d**,**g**), with characteristic large refractory inclusions (**b**–**g**), the size of which often exceeds the thickness of the cells (**c**,**d**,**e**,**g**).

**Figure 6 microorganisms-10-01172-f006:**
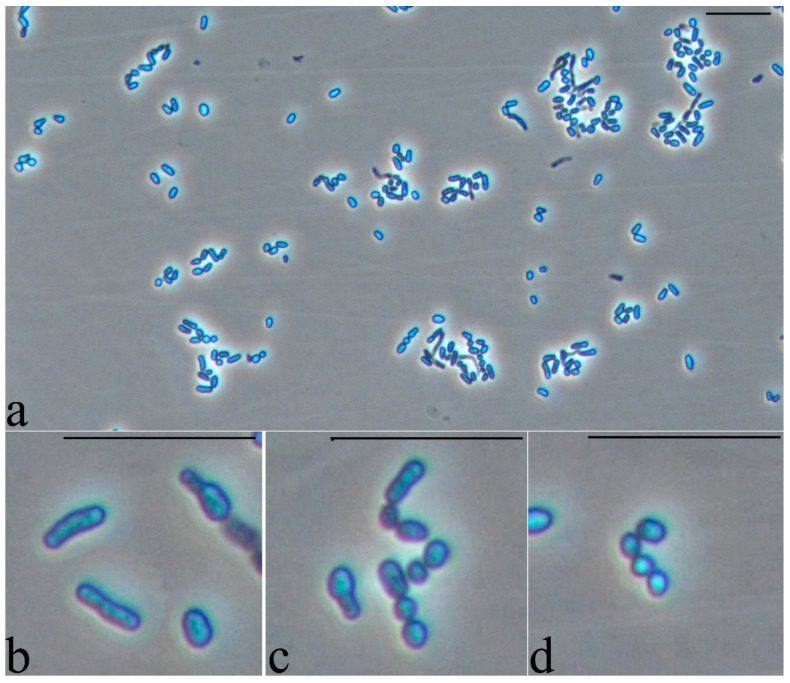
Strain S8 on glucose 4 days of growth. Light microscopy, phase contrast. The length of the scale bar is 10 µm. Crushing of rod-shaped cells occurs with the formation of chains of small cocci (**a**,**c**,**d**). Formation of multiple partitions inside the rod-shaped cell at the initial stages of crushing (**b**).

**Figure 7 microorganisms-10-01172-f007:**
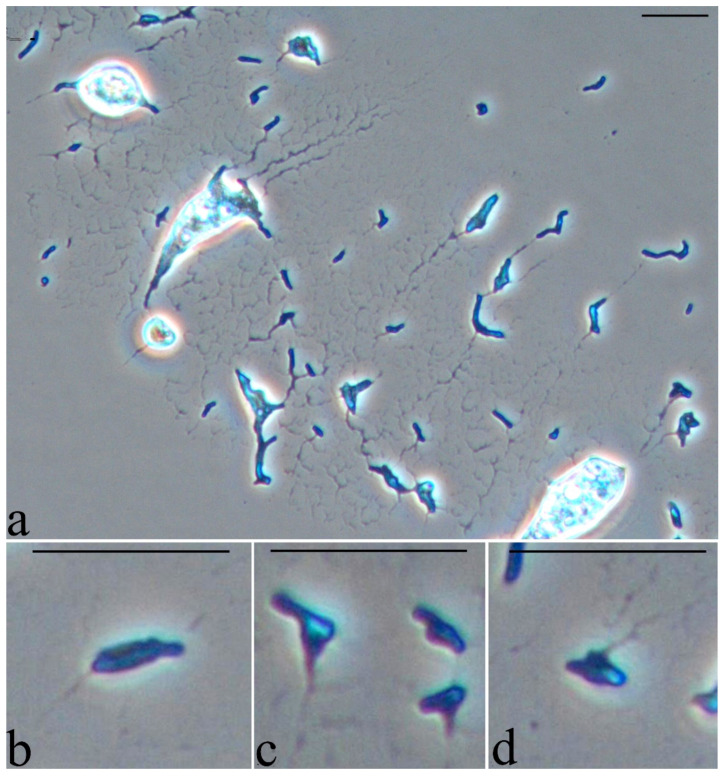
Strain S8 on hexadecane 4 days of growth. Light microscopy, phase contrast. The length of the scale bar is 10 µm. Bacterial cells are “pushed out” from the growth medium on hexadecane during preparation for light microscopy (applying the surface foam material, where the microbial utilization of hexadecane occurs, on a glass slide and pressing the drop with a cover slip) (**a**). Various forms of some cells with substrate residues on their surface (**b**–**d**).

**Figure 8 microorganisms-10-01172-f008:**
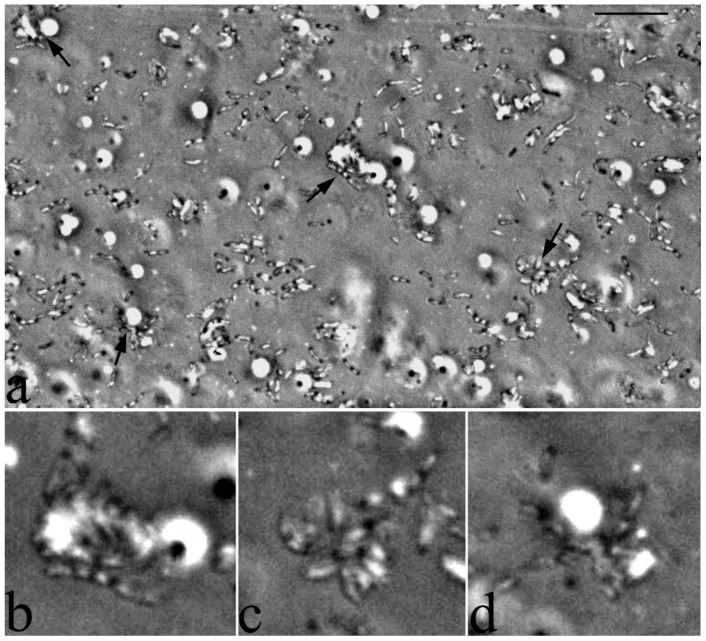
Strain S8 on hexadecane 4 days of growth. Bacterial microconglomerates formed around a certain central core with refractory properties. Indicated by arrows (**a**) and on enlarged fragments (**b**–**d**). Light microscopy, phase contrast. The length of the scale mark (**a**) is 10 µm.

**Figure 9 microorganisms-10-01172-f009:**
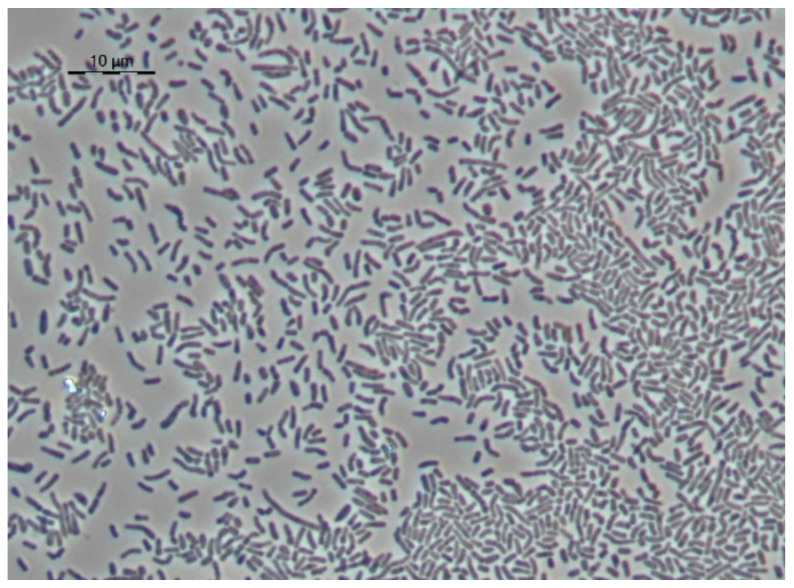
*R. qingshengii* strain VT6 on LB agar medium 1 month of growth. Light microscopy, phase contrast. The length of the scale mark is 10 µm.

**Figure 10 microorganisms-10-01172-f010:**
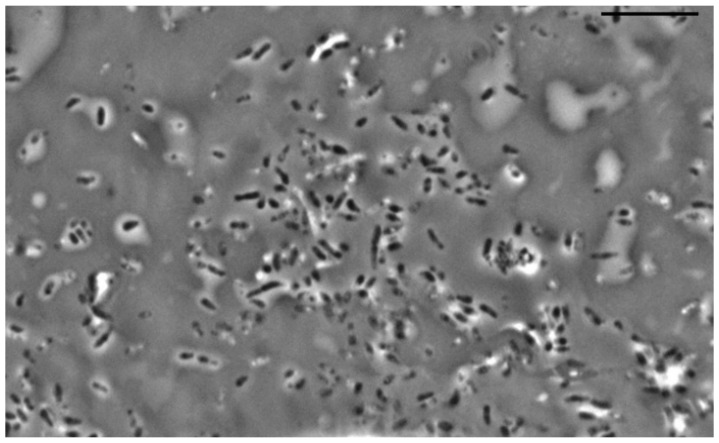
*R. qingshengii* strain VT6 on hexadecane 4 days of growth. Light microscopy, phase contrast. The length of the scale mark is 10 µm.

**Figure 11 microorganisms-10-01172-f011:**
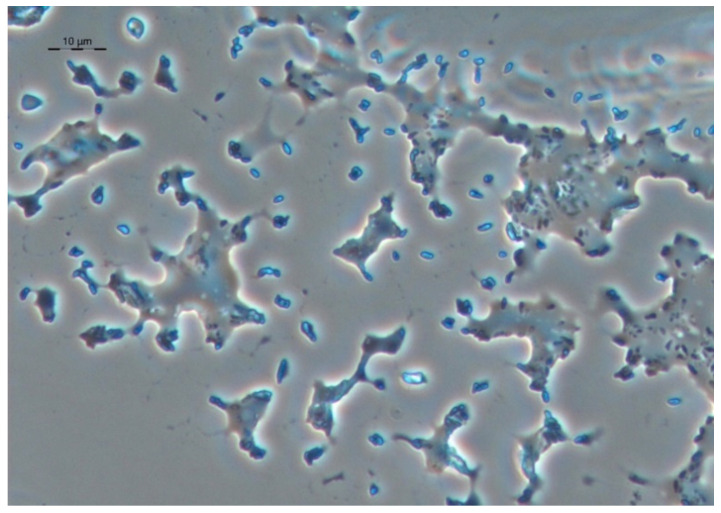
*R. qingshengii* strain VT6 in a drop on hexadecane 4 days of growth. Light microscopy, phase contrast. The length of the scale mark is 10 µm.

**Figure 12 microorganisms-10-01172-f012:**
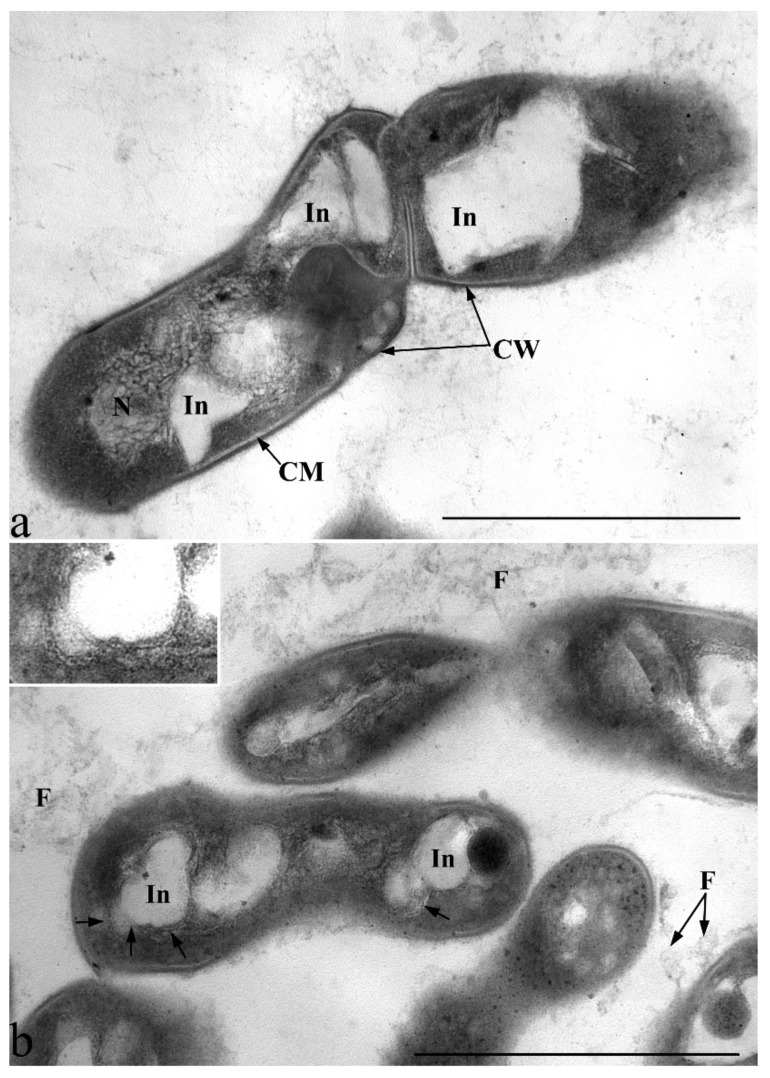
Ultrastructural organization of cells of strain S8 during growth on glucose (exponential growth phase) (**a**) and under growth conditions on dodecane (4 days) (**b**). Transmission electron microscopy. Ultrathin sections. Designations: N-nucleoid; CM—cytoplasmic membrane; CW—cell wall; In-electron-transparent inclusions; F—fibrillar surface structures. Arrows show membrane-like structures, which are also shown in the enlarged fragment in the inset in (**b**). Bar—1 µm.

**Table 1 microorganisms-10-01172-t001:** Information on primers for genes involved in surfactant biosynthesis in *R. opacus* S8 strain.

Target Gene	Primers	Primer Sequence	Amplicon Size, bp
16S rRNA	s8_16s_rt_for	GGACGAAGCGAAAGTGACG	130
	s8_16s_rt_rev	CGACAAACCGCCTACGA	
Alkane 1-Monooxygenase 1	s8_alkb2884_rt_for	CATCAACACCGCACACGAGA	213
	s8_alkb2884_rt_rev	TTCAGGCTTCCCCAGACACT	
Trehalose 2-Sulfotransferase	s8_tst1278_rt_for	TACCTGCCGTCCACCAATC	183
	s8_tst1278_rt_rev	CTTTCCGCCCCACACCC	
Trehalose-2-Sulfate Acyltransferase PapA2	s8_tsat7567_rt_for	GAAGCGGACCTGACGAAAGA	150
	s8_tsat7567_rt_rev	CGAGAAGGGGGCGGACA	

**Table 2 microorganisms-10-01172-t002:** Surface tension of the culture liquid of carbohydrate-oxidizing strains.

Strain	ST, mN/m
*Rhodococcus* sp. Ac2845	69 ± 1.1
*Rhodococcus* sp. P13	47 ± 1.2
*Rhodococcus* sp. P1	48 ± 1.1
*Rhodococcus* sp. G172	49 ± 1.3
*Rhodococcus* sp. 557	65 ± 1.6
*Rhodococcus opacus* S8	39 ± 0.8
*Rhodococcus ruber* P25	61 ± 1.4
*Rhodococcus* sp. P12	47 ± 1.2
*Rhodococcus* sp. VT7	52 ± 0.9
*Rhodococcus wratislaviensis* G10	52 ± 1.0
*Rhodococcus* sp. 412	54 ± 1.3
*Rhodococcus qingshengii* VT6	32 ± 0.7

**Table 3 microorganisms-10-01172-t003:** Comparison of *R. opacus* genomes deposited in the Genbank database. For comparison, only complete genomes were taken.

Strain	Genome Size, Mb	Chromosome Size, Mb	Plasmid Size, Mb
S8	8.78	7.75	788 135 105
R7 (GCA_000736435.1)	10.12	8.46	656 426 352 191 25
XM24D (GCA_012931745.1)	9.08	7.85, 1.23	-
KT112-7 (GCA_017916235.1)	8.00	7.58	282 131
1CP (GCA_001685605.1)	8.64	7.68	64 885
B4 (GCA_000010805.1)	8.83	7.91	111 4 558 2.7 245
PD630 (GCA_020542785.1)	9.17	8.37	538 168 85
DSM44186 (GCA_019856255.1)	8.84	8.04	744 52

## Data Availability

Data is contained within the article.

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
