# Peer review of "Complete Genome Analysis of Rhodococcus opacus S8 Capable of Degrading Alkanes and Producing Biosurfactant Reveals Its Genetic Adaptation for Crude Oil Decomposition"

_microorganisms, 2022, doi:10.3390/microorganisms10061172_

Round 1

Reviewer 1 Report

This manuscript investigates the genome of Rhodococcus opacus S8 and explores behavior traits of this strain grown in the presence of hexadecane.

1) What is the hypothesis of the research?

2) Results of the section "3.5. Study of the cell morphology when interacting with a hydrophobic substrate" should be compared with those of the published papers.

Author Response

The authors express their sincere gratitude to the reviewer for careful reading of the article and comments. As for the first remark.

1) What is the hypothesis of the research?

Unfortunately, or fortunately, to date, a huge amount of information has been accumulated regarding both the study of the genomes of rhodococci and the ability of these bacteria to decompose crude oil and its individual components. Starting work on the complete genome sequencing of R. opacus we did not expect to find anything completely new.

However, a number of issues remain insufficiently studied. For example, the details of the interaction of a cell with a hydrophobic substrate. For a long time we have been studying the response of a cell to a substrate at the cytological level, and all the time we are getting some new information that is not described or poorly represented in the literature. Therefore, by initiating this work, we hoped to obtain new results not so much on the structure of the genome, but on the structure of cells. And we hope that in this work we have presented the results obtained.

2) Results of the section "3.5. Study of the cell morphology when interacting with a hydrophobic substrate" should be compared with those of the published papers.

We have added to the discussion some articles that we could find that, in our opinion, are useful in terms of cytological research. If the reviewer knows some other articles on the study of cell cytology, we will be grateful for advice, but as far as we were able to track the literature data, there are no completely similar articles.

Reviewer 2 Report

Reviewer’s comments

The authors of the manuscript entitled “Complete genome analysis of Rhodococcus opacus S8 capable 2 of degrading alkanes and producing biosurfactant reveals its 3 genetic adaptation for crude oil decomposition “ have done an interesting and novel work deserving commendation. However, it lacks a deeper discussion, thus I strongly suggest addition of an deeper discussion and comparison with previous studies and the novelty of their work should be discussed. After having seen the novelty of their work and assuming that they would address my comments I suggest that their paper be accepted after minor revision.

Author Response

Answer to Reviewer 2.

The authors of the manuscript entitled “Complete genome analysis of Rhodococcus opacus S8 capable 2 of degrading alkanes and producing biosurfactant reveals its 3 genetic adaptation for crude oil decomposition “ have done an interesting and novel work deserving commendation. However, it lacks a deeper discussion, thus I strongly suggest addition of an deeper discussion and comparison with previous studies and the novelty of their work should be discussed. After having seen the novelty of their work and assuming that they would address my comments I suggest that their paper be accepted after minor revision.

Many thanks to the reviewer for the careful reading of the article and the comment made. In our opinion, the most interesting part of our work is the conducted cytological studies. Methodically, this is a very difficult part, since in order to understand a holistic and real picture of the interaction between the cell and the substrate, we had to obtain cell preparations that would not be spoiled during fixation and obtaining ultrathin sections.

We hope that we have succeeded in obtaining results on the basis of which we can formulate a new hypothesis for the provision of oxygen to cells during the interaction of a cell and a hydrophobic substrate. The genetic studies carried out to some extent confirmed the right of this hypothesis to exist. This concerns the detection of genes encoding truncated hemoglobin. We have added articles to discuss our hypothesis. Unfortunately, we did not find articles that would describe the features of cell cytology during interaction with a hydrophobic substrate. If the reviewer has such articles, please provide links. We would add them to the discussion too.

With gratitude, authors
